# Effect of Low-Frequency Electromagnetic Casting on Micro-Structure and Macro-Segregation of 5A90 Alloy Ingots

**DOI:** 10.3390/ma13122720

**Published:** 2020-06-15

**Authors:** Fuyue Wang, Xiangjie Wang, Jianzhong Cui

**Affiliations:** 1Key Lab of Electromagnetic Processing of Materials, Ministry of Education, Northeastern University, Shenyang 110819, China; jzcui@mail.neu.edu.cn; 2College of Materials Science and Engineering, Northeastern University, 314 Mailbox, Shenyang 110819, China

**Keywords:** 5A90, aluminum–lithium alloy, low-frequency electromagnetic casting, direct chill casting, micro-structure, macro-segregation

## Abstract

The effect of low-frequency electromagnetic fields on the micro-structure and macro-segregation of 5A90 alloy ingots during the semi-continuous casting process were quantitatively investigated. The ingots of a 5A90 alloy with a diameter 170 mm were produced by the conventional direct chill casting (DCC) process and low-frequency electromagnetic casting (LFEC) with 10 Hz/100 A. The results showed that LFEC can substantially refine the micro-structure and shorten the width of the columnar grain area of an ingot. The refinement effect came with the relieving of grain boundary segregation and an improvement in the macro-segregation of the ingot. Compared with the traditional DCC process, the tensile properties of the aged alloy prepared by the LFEC process were improved due to the effects of the increase in solid solubility and the strengthening of the grain refinement, so that the stability of the tensile properties was also improved. Meanwhile, the rate of yield increased by 2.3% with a decrease in the peeling thickness of the ingot.

## 1. Introduction

Al-Mg-Li alloys have been widely applied in aeronautics and spaceflight due to their outstanding comprehensive performance in terms of their low density, medium strength and high elastic modulus [1,2,3]. However, they are liable to cause some casting defects including coarse grain, severe dendrite structure (micro-segregation) and center line macro-segregation. In particular, the macro-segregation of large sized ingots produced by conventional direct chill casting (DCC) processes is serious [4], owing to the broad crystallization temperature interval of the alloy and the unsuitable heat transfer condition at the solidification front during casting [5], which not only reduces the yield rate of the alloy but also causes some adverse effects on its micro-structures and mechanical properties. Macro-segregation refers to the non-homogeneous distribution of alloying elements over a larger length scale, while micro-segregation is the in-homogeneity of the chemical composition on the scale of a single grain or dendrite. What makes macro-segregation acutely troublesome is that it cannot be eliminated by downstream processing, unlike micro-segregation, which could be relieved relatively easily by homogenization treatment. Therefore, it becomes rather important to minimize the effects of macro-segregation.

Over the last century or so, the mechanisms of macro-segregation in direct chill cast alloys have been already widely discussed. The main reason for macro-segregation is the partitioning of the alloying elements and the relative movement of the liquid and solid phases during solidification, whereby the advancing solidification front pushes the solute enriched liquid (in the case of distribution coefficient K < 1) toward the hotter part of the molten liquid, eventually leading the center of the casting to contain more solute than the periphery of the casting. This form of macro-segregation is termed normal segregation. However, the usually detected macro-segregation situation in direct chill cast aluminum alloy ingots is just the reverse. The center position of the ingots is solute lean while the peripheral position of the ingots is solute enriched, along with a region of negative subsurface segregation [6]. We know that the cooling intensity of melt at the border of sump is greater than that at the center of sump [7]. The grains at the center position of the ingot grow sufficiently under this relatively weak cooling intensity, and the solidified structure discharges the solute elements towards the liquid phase; meanwhile, solute-rich liquid is transported to solidification by solidification shrinkage [8]. The phenomenon of peripheral strong segregation could be imagined in terms of the solute-rich low melting point of the eutectic remelting and being squeezed out of the shell of the ingot by a strong solidification compressive strain when a certain position of the ingot is between the graphite ring and the second cooling water and solidifies at the surface as it exudates, which makes it extremely easy to cause over-burning during the subsequent heating treatment. It is worth noting that the optimized solidification condition and control of the melt flow at the solidification front would be an effective way to improve macro-segregation. At present, an effective and advanced semi-continuous casting process, low-frequency electromagnetic casting (LFEC) as developed by Cui [9], has attracted much attention in the field of aluminum alloy casting because of a number of advantages, such as refined micro-structure, improved surface quality of the ingot, elimination of central cracks and alleviation of macro-segregation. The LFEC process is based on the casting, refining and electromagnetic process (CREM) proposed by Vives [10]. The flow of melt in the sump is modified and the heat transfer is intensified by electromagnetic stirring. The relative movement of the liquid and solid phases changes significantly. Despite some literature being available about the effect of electromagnetic fields on the macro-segregation of aluminum alloy ingots, the relationship between micro-structure, alloying element solid solubility and macro-segregation has not yet been examined.

In this study, a low-frequency electromagnetic field was used for casting 5A90 Al-Li alloy. For comparison, the direct chill casting method was employed as well. The effect of LFEC on the micro-structure, micro-segregation and macro-segregation of 5A90 alloy were investigated and the related mechanism was discussed.

## 2. Experimental Materials and Methods

### 2.1. Material Preparation

The nominal compositions of 5A90 alloy used in this work were based on the chemical composition with Al-5Mg-2Li-0.1Zr (in wt.%). The raw materials used for melting were as follows: Al ingot (99.9%), Mg ingot (99.9%), Li ingot (99.85%), Al-5%Zr and A1-5%Ti-1%B master alloys as raw materials. They were smelted in an induction furnace with a 304 stainless crucible under the protection of argon, and a molten covering agent (50 wt.% LiCl and 50 wt.% LiF) was employed in order to prevent oxidation and reduce the loss of Li. The molten alloy was kept at 695 °C for 15 min before being moved from the furnace into the crystallizer mold through the TC4 titanium alloy launder. Finally, the ingot with a diameter of 166 mm was cast at a cooling water flow of 35 L/min and at a casting speed of 90 mm/min. The secondary cooling water was ejected at the surface of the ingot through the blowhole as soon as the ingots were removed from the crystallizer. When implementing LFEC, the LFEC device had already been started at the initial stage of casting. The experimental device of the LFEC is shown in Figure 1. The LFEC crystallizer included a water tank shell, a copper induction coil and a crystallizer mold with an inner diameter of 170 mm. The low-frequency electromagnetic field was induced by an alternating current with 10 Hz and 100 A with a 100-turn pure copper coil that surrounded the crystallizer. When implementing conventional DCC, the low-frequency alternating current generator was turned off while the other casting parameters remained the same. The ingots were homogenized with heat treatment at 480 °C for 36 h. The surfaces of the ingots were removed and then extruded at 380 °C into bars with diameters of 20 mm at an extrusion ration of 15. These bars were solid solution heat treated at 470 °C for 40 min and then quenched into cold water immediately. Finally, they were aging heat treated at 135 °C for 24 h.

### 2.2. Microscopy and Characterization

The samples for micro-structure observation were taken from different positions in the cross section of the ingot. The micro-structure of the as-cast structure was observed by a Leica DMI-5000M optical microscope (Wetzlar, Germany) with cross-polarized light. The morphology and elemental composition of the phases in the as-cast alloy were characterized on a SSX-550 scanning electron microscope (Herborn, Germany) equipped with a X-ray spectroscope. The area fraction of the eutectic structure in the SEM images was calculated by Image-Pro Plus (6.0, Media Cybernetics, Rockville, MD, USA). In addition, the JXA-8530 emission electron probe micro analyzer (JEOL, Tokyo, Japan) was used to quantitatively analyze the main alloying elements’ content. The chemical composition of the different positions of the 5A90 alloy ingots was detected using inductively coupled plasma-atomic emission spectrometry. The hardness test was carried out on a KB-SA hardness tester (KB, Bergheim, Germany). The hardness measurement was operated five times for each position to ensure the reliability of the data. The tensile measurement was carried out on an AG-Xplus electron testing machine (Shimadzu, Kyoto, Japan) at a strain rate of 2.0 mm/min at room temperature. The as-aged alloy bars were machined into standard tensile specimens according to ASTM-B557M. The tensile measurement was operated five times, and then the average values of the yield strength (YS), ultimate tensile strength (UTS) and elongation (EL) were calculated, respectively.

## 3. Experimental Results

### 3.1. As-Cast Optical Microstructure

Figure 2 demonstrates the micro-structure of the ingots cast by DCC and LFEC with 10 Hz/100 A. In general, the micro-structure of the DCC ingot mainly consisted of coarse rose-like dendrite grains with a larger size and showed high in-homogeneity with an increase in grain size from the border to the center of the ingot. The border of the ingot had a wide dendritic crystal area with a width of approximately 2.8 mm. With the application of the LFEC process, the micro-structure became smaller and more uniform, while the refinement effect of LFEC at the center was slightly weaker than that at the border. The grain morphology was translated from rose-like dendrite grains to fine equiaxed grains under the influence of the LFEC. It is also notable that the width of the dendritic crystal area was reduced to less than 2.0 mm. To understand the refinement effect, the average grain size was measured. Figure 3 shows the grain size distribution from the surface to the center. It could be obviously found that the grain size of the LFEC ingot was smaller than that of the DCC ingot and the difference of the grain size in the LFEC ingot along the radius was smaller as well. For the DCC ingot, the grain size at the border was about 50 μm and at the center was close to 70 μm. In comparison, the grain size at the border of the LFEC ingot was about 40 μm and at the center was less than 50 μm. Therefore, the LFEC process could significantly refine the grain and shorten the width of the columnar grain area of ingots.

### 3.2. Macro-Segregation

The concentration of alloying elements at different positions of the 5A90 alloy ingots was detected using ICP-AES, and the degree of segregation can be expressed as the relative segregation rate as shown in Equation (1).
(1)$$\Delta C=\frac{c_{i}-c_{o}}{c_{o}}$$
where *c_i_* is the concentration of alloying elements detected at each position, and *c_o_* is the initial concentration of alloying elements. Figure 4 shows the relative segregation rate of the major alloying elements (Mg and Li) versus different positions of the 5A90 ingots. It could be seen from the value of the relative segregation rate that the Mg exhibited a higher segregation tendency as compared to Li. This is due to the different equilibrium distribution coefficients of Mg and Li in aluminum (k_Mg_ < k_Li_). There is the large segregation rate of Mg, which appeared close to the surface of the ingot, with a maximal relative segregation rate of 1.20%. The ingots prepared by DCC and LFEC had a nearly identical segregation pattern. The negative segregation occurred in the center and strong positive segregation occurred on the ingot’s surface, accompanied by a strong negative segregation zone close to the surface. However, the degree of segregation was obviously alleviated by LFEC through the whole cross section from surface to center. Depending on the results above, LFEC can effectively improve alloying element macro-segregation of the 5A90 alloy ingot and narrow down the strong negative segregation zone which occurs near the surface of the ingot.

### 3.3. Hardness

In order to further study the effect of macro-segregation on the hardness of ingots under DCC and LFEC, the hardness of the as-cast and as-solution alloys was measured by a Vickers hardness tester. Figure 5 shows the Vickers hardness distribution of 5A90 alloy along the radial direction. It is evident from the figure that the hardness of the LFEC alloy was higher than that of the DCC alloy in both as-cast and as-solution forms. The hardness of the DCC ingots decreased along the radial direction from the border to the center. However, with the application of LFEC, the hardness difference along the radial direction clearly decreased, and the hardness tended to be more uniform. The increment of hardness in the as-cast alloy can be attributed to grain refining and solution strengthening, which is consistent with the experimental results shown in Figure 3 and Figure 4. The results reveal that the hardness of ingots along the radial direction is strongly affected by LFEC, and this effect is carried from the casting stage to the solution treatment stage.

### 3.4. Micro-Segregation

Figure 6 shows the morphology of the phases of the as-cast 5A90 alloy in the different positions. In both DCC and LFEC alloys, there were three kinds of structure distributed in the Al matrix. According to the EDS results, as shown in Figure 7, it could be speculated that the point A is an Mg-rich eutectic phase, point B is an Fe-rich inter-metallic compound and point C is a primary Al_3_Zr particle. The primary Al_3_Zr particles presented square and rhombus shapes, and the size generally did not exceed 30 μm. Some of the primary Al_3_Zr particles existed in the center of the grain as heterogeneous nuclei, and the other coarse primary particles got together and distributed at grain boundaries. It was reported the coarse primary Al_3_Zr particle would be very difficult to dissolve into the matrix during the homogenizing [11]. It can also be observed from Figure 6 that the laminar eutectic phase in the DCC alloy was coarse and very uneven. After the application of LFEC, the morphology of the fine eutectic phase replaced the coarse laminar eutectic phase.

During the non-equilibrium solidification process, the solidified grain of the solidification front continuously discharges the alloying elements into the front of the solid/liquid interface. At the late stage of solidification, a large amount of alloying elements accumulate in the inter-dendritic residual liquid, and then a non-equilibrium eutectic is formed at the grain boundary. The fraction of eutectic in the grain boundary can be used as an indicator of micro-segregation. In our study, the area fraction of eutectic in the DCC and LFEC as-cast alloys in the SEM images were accounted for by Image-Pro soft, as shown in Figure 8. The area fraction of eutectic in the DCC alloy decreased from 4.11% in the border of the ingot to 3.75% in the 1/2 radius of the ingot and 3.5% in the center of the ingot. Meanwhile, in the LFEC alloy, the area fraction of eutectic from the ingot’s border to the center of the ingot was 3.57%, 3.30% and 3.17%, respectively. Under the same position of the ingot, the eutectic of the DCC alloy was much greater than that of the LFEC alloy. This indicates that the LFEC can inhibit the formation of coarse Mg-rich eutectic.

To further investigate the micro-segregation occurring in the micro-structure of DCC alloys and LFEC alloys, electron probe micro analyzer was carried out in order to quantitatively analyze the concentration of main alloying elements at the 1/2 radius of the as-cast alloy. Respectively, along the lines across the coarse dendritic grains and equiaxed grains in Figure 9a,c, the line scan results of the Mg element as detected by EPMA are illustrated in Figure 10. It can be seen from Figure 9 that the Mg element was segregated along the grain boundary, forming coarse eutectic phases and presenting a dendrite profile in the DCC alloy. By contrast, with the grains being refined and eutectic phases becoming finer, the distribution of the Mg element showed a higher uniformity in the LFEC alloy. Line scan measurements of the DCC alloy represent a considerable difference in the concentration of Mg between the interior of the grain and the grain boundary, while the fluctuation of the line scan measurements of the LFEC alloy is relatively small. In this work, the element distribution uniformity parameter is utilized to judge the degree of micro-segregation, average concentration of Mg in the scan line (A), standard deviation of concentration of Mg in the scan line (S) and distribution uniformity parameter of Mg element (ξ), which are calculated as shown in Equation (2).
(2)$$\left\{ \begin{array}{l} A=\sum_{i=1}^{n}x_{i}/n\\ S=\sqrt{\sum_{i=1}^{n}(x_{i}-A{)}^{2}/n}\\ \xi =S/A \end{array}\right.$$
where *x_i_* is the concentration of Mg at position *i*. According to the results, the average concentration and distribution uniformity of Mg in the scan line of the LFEC alloy all outperformed the DCC alloy. It can be concluded that the concentration of Mg in the interior of the grain increases greatly and the segregation along grain boundaries is effectively inhibited after applying the LFEC.

### 3.5. Product Performance and Rate of Yield

To illustrate the refinement effect of LFEC on the micro-structure, the micro-structures of the 5A90 DCC and LFEC as-aged alloys are shown in Figure 11. These are composed of strip-like grains arranged along the extrusion direction and some second-phase particles distributed along the grain boundaries. The average width of the strip-like grains decreases from 27.5 μm in DCC to 19.2 μm in LFEC. This fully demonstrates that the refinement effect of LFEC on grain size and morphology could be transmitted from the casting stage to the aging stage.

Figure 12 demonstrates the tensile properties of the 5A90 DCC and LFEC as-aged alloys. After applying the LFEC, the average ultimate tensile strength and yield strength of the as-aged alloys increased by 40 MPa and 33 MPa, respectively. The elongation could be considered to remain invariable. The improvement in tensile properties could mainly be attributed to the effect of solid solution strengthening and grain refinement strengthening. It is worth noting that the standard deviation of the strength and elongation of the alloy significantly decreased after applying the LFEC. This suggests that the stability of the tensile properties of the LFEC alloy could be improved by suppressing macro-segregation. In addition, the variation of yield as a function of peeling thickness in the different diameters of the ingot is shown in Figure 13. According to the calculations, the rate of yield increases with a decrease in peeling thickness. Based on the experimental results presented in Section 3.1 and Section 3.2, LFEC could significantly shorten the width of the columnar crystal area and the strong negative segregation area; thus, the peeling thickness could decrease from 5 mm to 4 mm. Due to this result, the rate of yield will increase by 2.3%. In short, applying LFEC can not only improve the stability of an alloy’s mechanical properties but also increase its rate of yield.

## 4. Discussion

### 4.1. Grain Refining

Many papers have discussed the refining of micro-structures in Al and Mg alloys by LFEC, which have exhibited the direct and remarkable function of LFEC [12,13]. The main reason for grain refinement during LFEC is the fast cooling effect due to the intensified heat transfer between the liquid and the wall of the mold and at the solidification front resulting from the fast flow that is enforced by the Lorentz force [14]. In this study, the fast cooling resulted in a deep overcooling which caused more Al_3_Zr particles to be activated and formed nuclei for new grains, and the flow and rotation restricted the grain growth, which resulted in the formation of fine, equiaxed grains in the ingot.

### 4.2. Decrease in Eutectics in LFEC Ingot

A new grain after nucleation is grown and a liquid/solid interface formed. The solutes in grains were discharged from grains to the liquid through the interface [15]. The amount of alloying elements that were solid solved in the solid grains increased and the amount that were discharged from grains to the liquid was reduced because the cooling rate of solidification was intensified and part of the energy needed for lattice distortion from the solute atom solid solved in the matrix was supplemented by LFEC [16], which resulted in an increase in the effective distribution coefficients of the alloying elements and the starting value of solutes meant the improvement of micro-segregation [17]. The most potential nuclei were activated by the application of the low-frequency electromagnetic field, which increased the amount of new grains and refined grains so that the amount of solutes discharged from grains to the liquid decreased because of a decrease in the small size of the grains. These results led to the low areal fraction and the small size of the non-equilibrium eutectics.

### 4.3. Improvement of Macro-Segregation

As we know, when the ingot drops to the bottom part of the mold, a gap will form between the wall of the mold and the solid shell of the ingot in DC casting because of the shrinkage from solidification. When the ingot moves down to the position below the mold and above the second cooling water spray, the ingot will be back-heated because the cooling rate decreases sharply, so that the liquid rich in alloying elements in the inner of the ingot will exudate to the surface region through the grain boundaries of the solid shell under the shrink force from solidification, which results in inverse segregation [6,8]. In this study, a layer with large sized dendritic grains formed in the border of the ingot because the grains in the solid shell which grew at the back-heating region resulted from the large distance of the back-heating region. The layer with large sized dendritic grains decreased and the inverse segregation was improved, which was due to the heat transfer between the wall of the crystallizer mold and the shell of the ingot was intensified by the application of a low-frequency electromagnetic field [18].

Negative segregation of Mg and Li elements occurred in the 5A90 alloy ingots. The “floating grains” theory is usually used to discuss negative segregation in DC casting [19]. The grains which solidified early were short of solute elements, and their density was higher than that of the liquid, so they moved along the solidification front and were deposited in the center of the sump; thus, the negative segregation of the alloying elements in the center of the ingot occurred. On the other hand, the shrinkage in the sub-surface forced the liquid full of solute elements to move between the dendrite networks and led to a more severe negative segregation in the sub-surface.

The mechanism of forced flow of melt under a low-frequency electromagnetic field during semi-continuous casting has been fully and extensively investigated. In the LFEC process, the melt is subjected to electromagnetic body forces. The Lorentz force consists of two parts, which are expressed as shown in Equation (3).
(3)$$F=J\times B=\frac{1}{\mu }\left(B\nabla \right)B-\frac{1}{2\mu }\nabla B^{2}$$
where *B* and *J* are the induced magnetic intensity and induced current generated in the melt, and *μ* is the permeability of the melt. The first term on the right hand of Equation (3) is a rotational component which results in the forced flow of melt. The second term is a potential force balanced by the static pressure of the melt, resulting in the formation of meniscus and a decrease in the contacting pressure on the crystallizer [20]. Under the effect of the vigorous forced flow induced by the Lorentz force, the temperature field becomes more homogeneous and the depth of the sump becomes shallower [21], which cannot be achieved during the DCC process; additionally, this effect could greatly increase the mixing of melt, since solutes at different positions on the solidification front are adequately exchanged [22]. The solidification conditions at the different locations of the sump tend to be consistent, while the difference in grain size along the radius of the ingot becomes minor [23]. For the structure at the edge of the ingot, the width of the coarse dendritic grain area of the ingot is obviously shortened under LFEC, and the melt can be decomposed into a potential force Fr and a rotational force Fz. Under this action of the potential force, the surface of the melt appears a meniscus, and the contacting pressure and contacting area on the mold can also be greatly reduced, which results in the growth of the columnar grain being inhibited. Instead, the fine equiaxed grains with zigzag grain boundaries are processed. As shown in Figure 14, the Lorentz force greatly increases the difficulty of exudation of the solute-rich eutectic that ultimately restrains the macro-segregation on the border of the ingot. The experimental results reveal that the refinement of the micro-structure by the LFEC process contributes significantly to the improvement of macro-segregation.

In addition, forced convection increases the under-cooling, which results in a larger number of nucleations taking place simultaneously. This means that not only that the micro-structure is refined and uniform, but the formation of coarse eutectic is inhibited and the segregation of alloying elements along grain boundaries is relieved [24]. This phenomenon must be accompanied by an increase in the solid solubility of the elements in the alloy. In the as-aged 5A90 alloy, the main strengthening precipitate is spherical δ′ (Al_3_Li), with a coherent orientation relationship with the Al matrix [25]. Due to the addition of Zr, recrystallization is effectively inhibited and the toughness is improved; this is mainly attributed to the formation of the β′ phase [26,27]. The β′ phase could act as a heterogeneous nucleation site for the δ′ phase during aging due to a decrease in the nucleation surface energy and strain energy of the δ′ phase [28]. The δ′ phase wraps the β′ phase, forming a β′/δ′ composite precipitate, which is hard to be sheared by dislocations and causes an obvious strengthening effect [29]. An increase in Mg and Li atoms within the grains enhances super-saturation in the matrix, which provides the necessary conditions for the precipitation of the strengthening phase [30].

## 5. Conclusions

In this work, the effects of a low-frequency electromagnetic field on micro-structure and macro-segregation in 5A90 Al-Li alloys during the casting process were quantitatively investigated. The major results were as follows:(1)The coarse rose-like grain is replaced by the fine equiaxed grain and the width of the dendritic crystal area becomes narrow under the influence of a low-frequency electromagnetic field.(2)The macro-segregation of the alloying elements (Mg and Li) of the 5A90 alloy’s ingot is effectively improved by the LFEC process, especially in the border of the ingot. The improvement effect of macro-segregation is transferred from the casting stage to the solution treatment stage.(3)The refinement effect of the grain is accompanied by the relieving of grain boundary segregation, and this effect comes with an improvement in the macro-segregation of the ingot.(4)Compared with the traditional DCC process, the tensile properties of the aged alloy prepared by the LFEC process is improved and the stability of the tensile properties is also improved. Meanwhile, the rate of yield increases by 2.3% with a decrease in the peeling thickness of the ingot.

## Figures and Tables

**Figure 1 materials-13-02720-f001:**
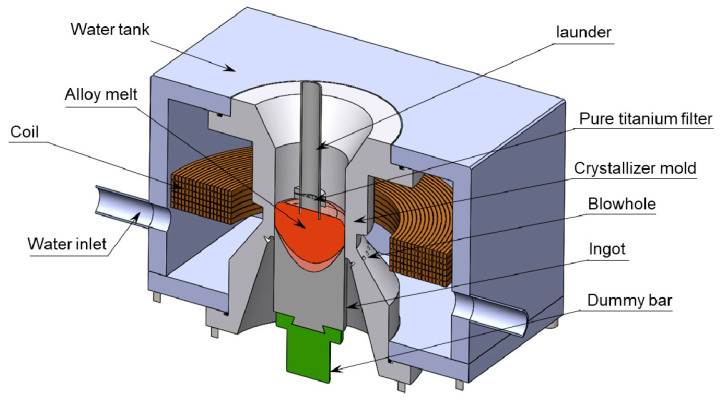
Schematic diagram of low-frequency electromagnetic casting experiment equipment.

**Figure 2 materials-13-02720-f002:**
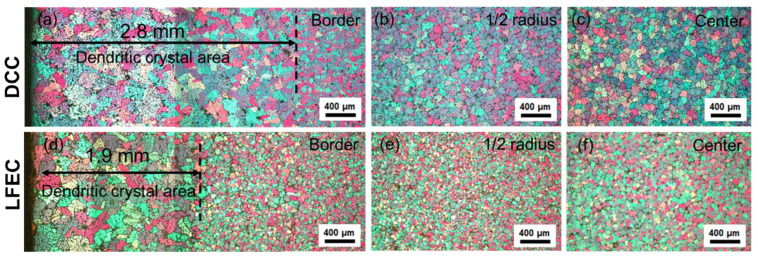
Micro-structure from border to center of as-cast 5A90 ingots: (**a**–**c**) direct chill casting (DCC) ingot; (**d**–**f**) low-frequency electromagnetic casting (LFEC) ingot.

**Figure 3 materials-13-02720-f003:**
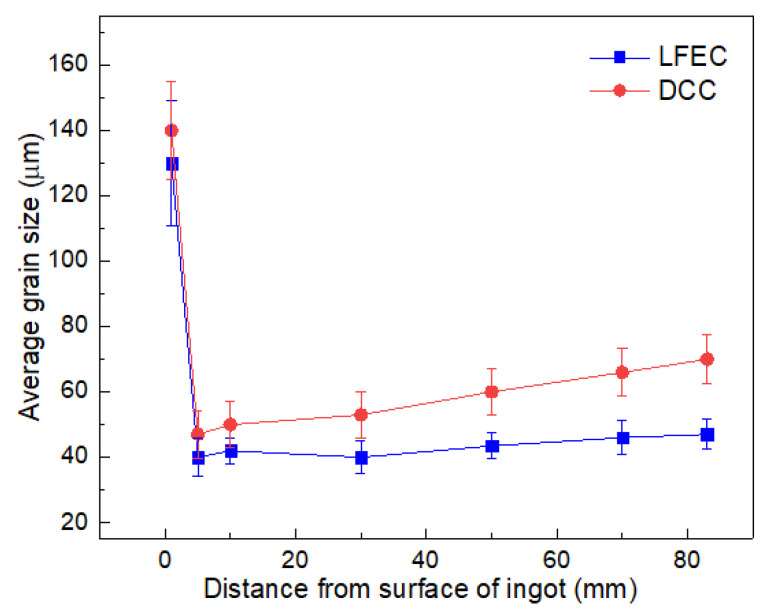
Statistic data of grain size of ingots.

**Figure 4 materials-13-02720-f004:**
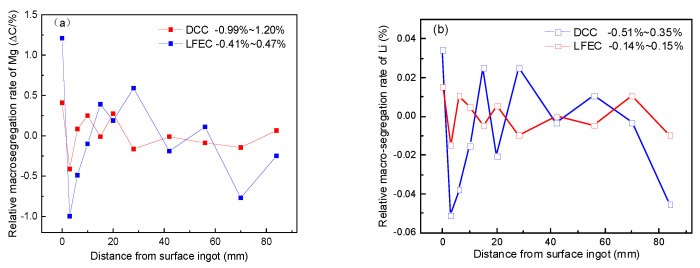
Relative segregation rate of alloying element of 5A90 ingots cast by DCC and LFEC with 10 Hz/100 A. (**a**) Mg; (**b**) Li.

**Figure 5 materials-13-02720-f005:**
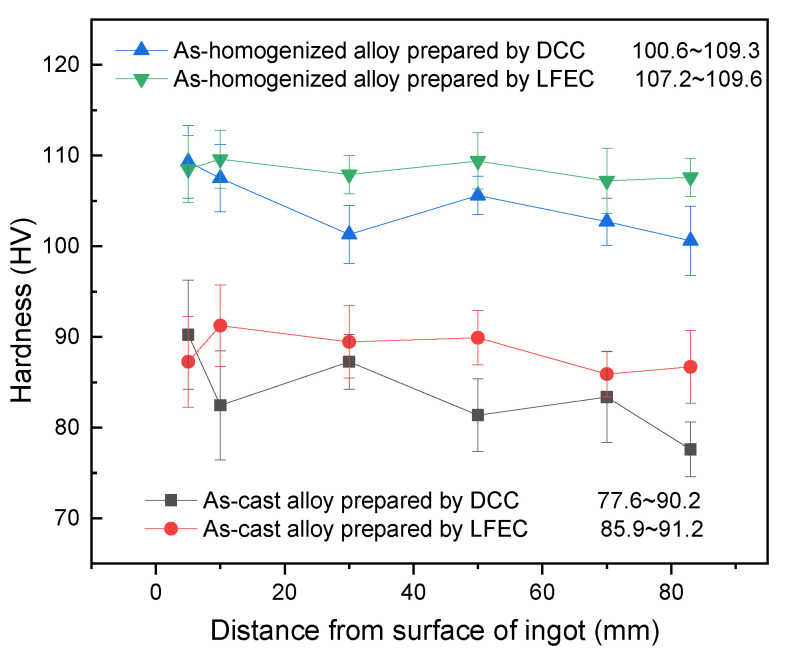
Hardness distribution of 5A90 alloy ingots along radial direction.

**Figure 6 materials-13-02720-f006:**
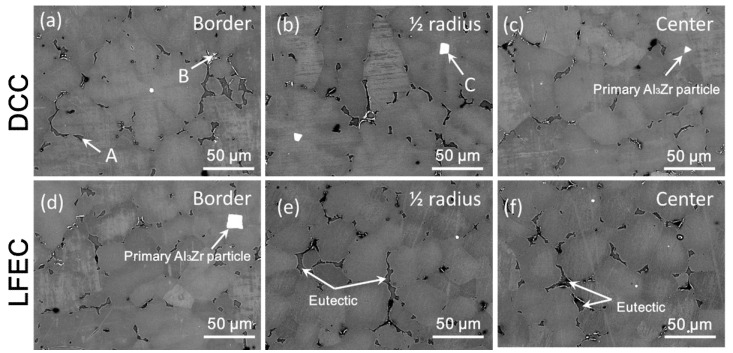
SEM images of 5A90 as-cast alloy. (**a**) DCC-border; (**b**) DCC-1/2 radius; (**c**) DCC-center; (**d**) LFEC-border; (**e**) LFEC-1/2 radius; (**f**) LFEC-center.

**Figure 7 materials-13-02720-f007:**
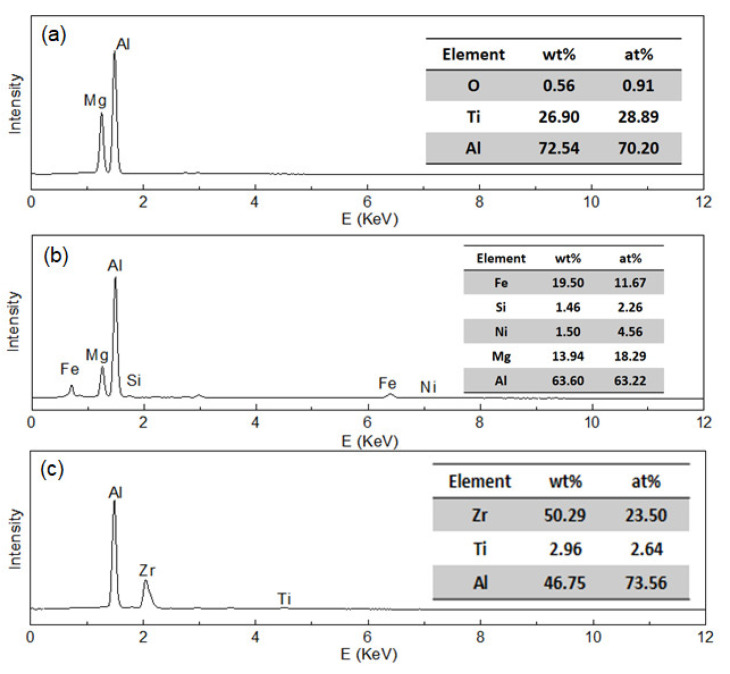
EDS results of 5A90 as-cast alloy in the SEM images. (**a**) Point A; (**b**) Point B; (**c**) Point C.

**Figure 8 materials-13-02720-f008:**
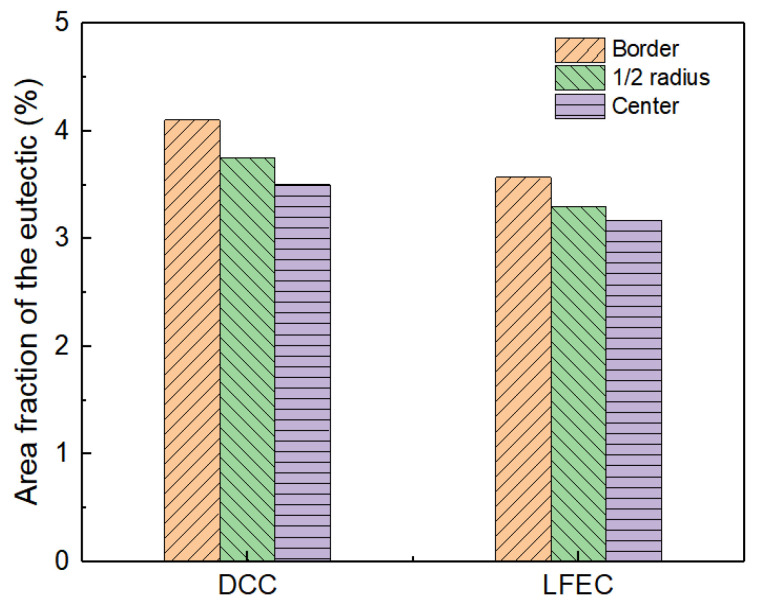
Area fraction of eutectic results of 5A90 as-cast alloy in the SEM images.

**Figure 9 materials-13-02720-f009:**
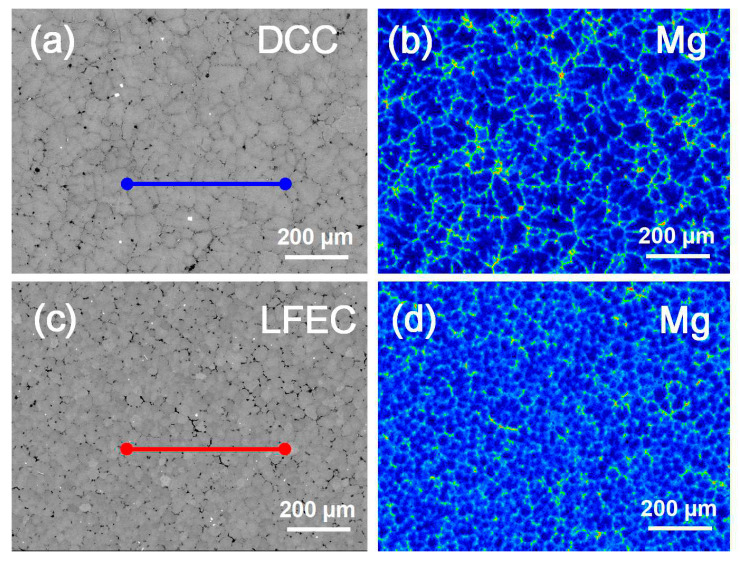
SEM images and Mg element mapping of 1/2 radius position of 5A90 as-cast alloy. (**a**) SEM image of DCC alloy, (**b**) Mg element mapping of DCC alloy, (**c**) SEM image of LFEC alloy, (**d**) Mg element mapping of LFEC alloy.

**Figure 10 materials-13-02720-f010:**
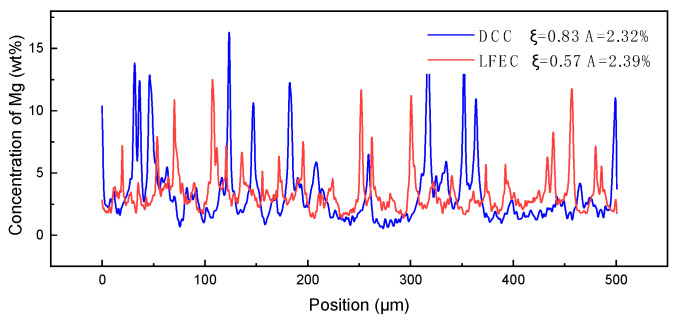
Concentration of Mg detected by EPMA along the line in Figure 9a,c.

**Figure 11 materials-13-02720-f011:**
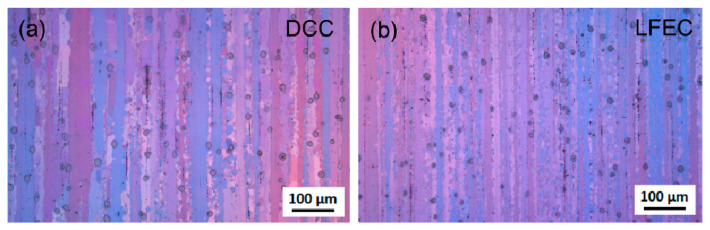
Micro-structure of 5A90 as-aged alloy: (**a**) DCC; (**b**) LFEC.

**Figure 12 materials-13-02720-f012:**
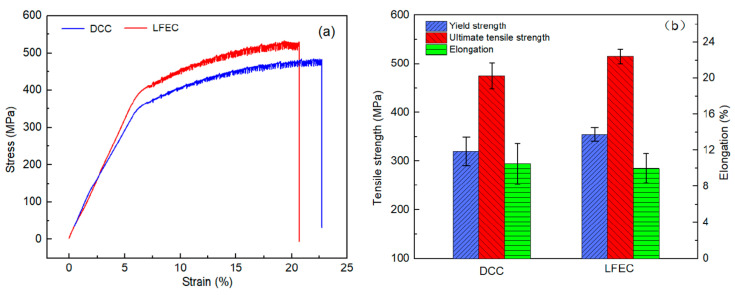
Mechanical properties of the 5A90 DCC and LFEC as-aged alloys. (**a**) engineering stress-strain curves; (**b**) tensile properties.

**Figure 13 materials-13-02720-f013:**
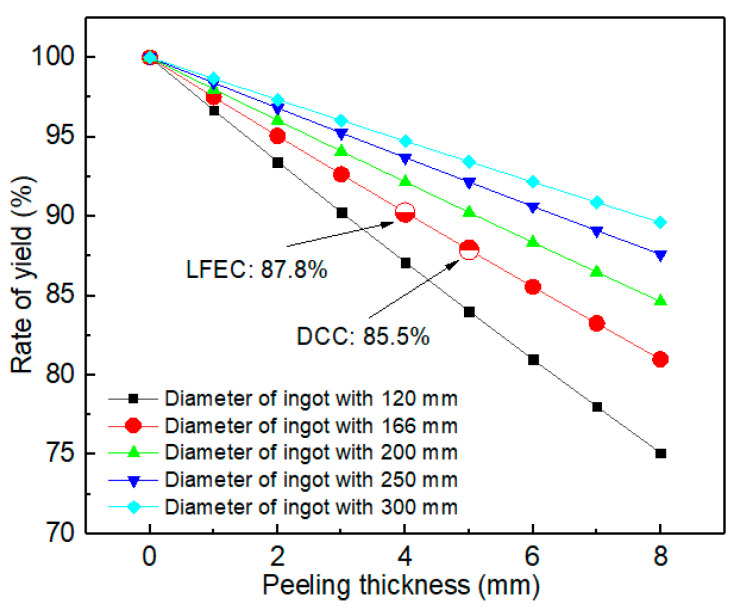
Variation of yield as a function of peeling thickness in the different diameters of the ingot.

**Figure 14 materials-13-02720-f014:**
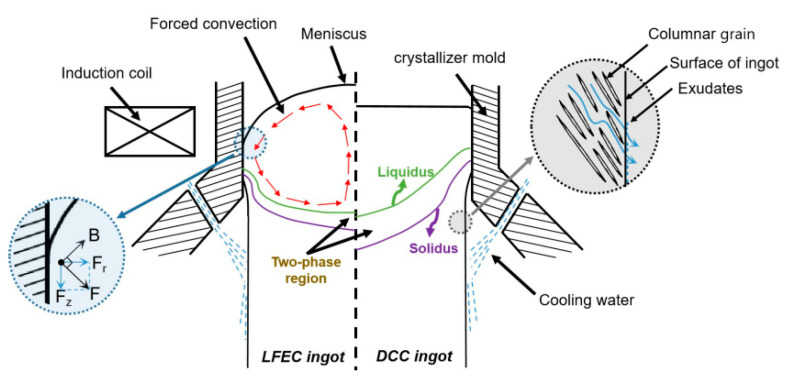
Schematic diagrams of the solidification.

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
