# Peer review of "Effect of Low-Frequency Electromagnetic Casting on Micro-Structure and Macro-Segregation of 5A90 Alloy Ingots"

_materials, 2020, doi:10.3390/ma13122720_

Round 1

Reviewer 1 Report

This article concerns to the effect of low-frequency electromagnetic casting on the microstructure, macro-segregation and tensile properties of 5A97 Al-Li alloy. The authors have carried out the melts in DCC and LFEC in order to establish a relationship between microstructure and the macro-segregation. The results are compared with the traditional DC casting process. The experimental results indicate that LFEC can significantly improve the microstructure and macro-segregation of ingots. This is a very interesting subject both to the scientific and industrial communities, and general findings are very promising. The technical contents suit the scope of Materials. It would be a good study that can guide the actual production of casting. However, there are still some contents that need to improve. Therefore, it is suggested that this paper may deserve publication after a modification. Following are a list of grammar errors and suggestions the authors may take into account to deeply review their article.

Introduction

  1. It is unable be eliminated by the downstream processing

-〉it is unable to be eliminated by the downstream processing

Experimental results:

  1. The results in Fig.5, Error bar should be given in the results of vickers hardness.
  2. can be decompose into      -〉can be decomposed into
  3. The title in Fig.10 shoule be revised.

Fig 10. Concentration of Mg detected by EPMA along the line in Fig. 8(a) and(c)

-〉 Fig. 10. Concentration of Mg detected by EPMA along the line in Fig. 9(a) and(c)

  1. Why do the authors separate inverse segregationand macro-segregation while writing the manuscript?It is recommended to put the two parts together.

Author Response

Reviewer: #1

  1. The comment(grammar error)

Response: We are sorry for the mistake in p1 line 39: "It is unable be eliminated by the downstream processing" has been replaced by "t is unable be to eliminated by the downstream processing"

  1. The comment(The results in Fig.5, Error bar should be given in the results of vickers hardness.)

Response: We appreciate for the reviewer’s valuable advice. We have added the error bars (standard error value) on the hardness plot.

  1. The comment(grammar error)

Response: We are sorry for the mistake in p12 line 318: "... can be decompose into..." has been replaced by "... can be decomposed into..."

  1. The comment(clerical error)

Response: We are sorry for the poor mistake.The title in Fig.10 has be revised.

P8 line 212: "Fig 10. Concentration of Mg detected by EPMA along the line in Fig. 8(a) and(c)" has been replaced by "Fig. 10. Concentration of Mg detected by EPMA along the line in Fig. 9(a) and(c)"

  1. The comment (Why do the authors separate inverse segregationand macro-segregation while writing the manuscript? It is recommended to put the two parts together.)

Response: It is quite right as the reviewer commented. The original manuscript has tried to discuss the inverse segregation and macro-segregation, which corresponds to two parts, the first is the 4.3 part, the second is the 4.4 part. However, the entire article is logically not compact in this way. We attempted to avoid this in the marked-up manuscript, as the reviewer advised. Therefore, we have merged parts 4.3 and 4.4 into one part to discuss macro-segregation.

We appreciate for editor Zhang and reviewers’ warm work , and hope that the correction will meet with approval. We will never give up the chance to improve the quality of our manuscript. 

Reviewer 2 Report

The authors present a thorough and clearly-presented manuscript on improving casting technology for Al-Mg-Si alloys. There are points where the English expression could be revised, but the thrust of the article remains clear.

Specific queries:

Introduction

Page 1 Line1: Corrosion resistance

The authors state that Al-Mg-Li alloys have excellent corrosion resistance. Can the authors justify this claim? Both Mg and Li are more reactive than Al and high corrosion resistance seems counter intuitive. Of the three references cited, two (Williams, J.C.; Starke, E.A. 1983 and Lavernia, E.J.; Grant, N.J.1987) did not support it, either.

Experimental

Page 4, figure 2: please enlarge scale bars

Page 5 Figure 4: Comment: I am not sure whether the curves shown here actually aid in the presentation of the data.

Page 6 Figure 5: Error bars (standard error) should be included on the hardness plot.

Page 7: " During the solidification process, the solid phase continuously discharges the alloying elements into the front of the solid-liquid interface."

Yes, there will be solid state diffusion to the solid-liquid interface, however this will be orders of magnitude slower than in the liquid phase.

Figure 10: Would it be possible to include upper and lower quartiles, or 95% bands etc. to show the overall difference between the two processes?

Figure 11: I suggest the authors consider adding representative tensile curves, possibly as supplementary material.

Author Response

Reviewer: #2

  1. 1. The comments(Corrosion resistance)

Response: Thanks a lot for the careful reviewer. We have revised the previous inaccurate statement and changed "corrosion resistance" to "high elastic modulus"

  1. The comments(Page 4, figure 2: please enlarge scale bars.)

Response: Thank you for your valuable advice. We have remade the figure and enlarged the scale bars (400 mm) in every picture to ensure the reader can see it clearly.

  1. The comments(Page 5 Figure 4: Comment: I am not sure whether the curves shown here actually aid in the presentation of the data.)

Response: It is quite right as the reviewer commented. The original manuscript has tried to present the macro-segregation trends by the curves. However, the curve does not pass through the real measurement point, so it cannot accurately represent the true chemical composition of each location. We attempted to avoid this in the marked-up manuscript, as the reviewer advised. Therefore, the curves are replaced by the polylines. In this way, it can not only accurately express but also present the entire macro-segregation trend.

  1. The comment(Error bars (standard error) should be included on the hardness plot)

Response: We appreciate for the reviewer’s valuable advice. We have added the error bars (standard error value) on the hardness plot.

5.The comment (Page 7: "During the solidification process, the solid phase continuously discharges the alloying elements into the front of the solid-liquid interface." Yes, there will be solid state diffusion to the solid-liquid interface, however this will be orders of magnitude slower than in the liquid phase.)

Response: We appreciate again for the reviewer’s commented. During non-equilibrium solidification, the solidified grains continuously discharge alloying elements towards liquid through the solid-liquid interfaces. The liquid containing a high concentration of alloying elements will remain inter-dendritic when dendrites contact. In traditional DC casting, due to poor fluidity in the solid-liquid two phase region, the solutes of the solidification front can not be adequately exchanged. The diffusion of solute element in the solid-liquid two phase region would be inhibited. To better express, we have replaced "solid phase" with "solidified grain of solidification front"

  1. The comment(Figure 10: Would it be possible to include upper and lower quartiles, or 95% bands etc. to show the overall difference between the two processes?)

Response: We appreciate again for the reviewer’s suggestion. We tried to quantitatively analyze the grain boundary segregation and the intragranular content of alloy elements, so we selected a line about 500 μm long for line scanning and the scan line crossed about a dozen grains. The results can explain the distribution of alloying elements within and outside the grains under DCC and LFEC, and the overall difference between the two processes can be seen from the element mapping by the bright colors.

  1. The comment(Figure 11: I suggest the authors consider adding representative tensile curves, possibly as supplementary material.)

Response: We appreciate again for the reviewer’s suggestion. We have added the tensile curves in Fig 12 as reviewer suggestion.

We appreciate for editor Zhang and reviewers’ warm work , and hope that the correction will meet with approval. We will never give up the chance to improve the quality of our manuscript. 

Reviewer 3 Report

Specific comments below:

  1. You didn’t explain, why such conditions for homogenization were selected: 480oC, 36h – the time is so long, the ingots were not big – diameter of 166mm. Probably the shorter time would be also effective and sufficient for ingots homogenization.
  2. Fig. 4 – it would be better when the relative segregation rate of Mg and Li will be presented on the graphs with the same scale – then it is easier to compare
  3. Line 145, page 5: ci and co should be a bottom line: ci, co
  4. Page 5 – please explain what is the reason of higher segregation tendency of Mg, than Li
  5. Fig. 5 – the standard deviation should be also presented on the graph of hardness distribution
  6. The microstructure performed with higher magnification should be presented in the paper (higher, than presented in Fig. 6 and 9) – then the morphology of the eutectic phase and also others phases will be visible and can be described.
  7. Why the authors presented only the microstructures of the ingots? Why you didn’t present the microstructure of the bars in T6 temper? – this is also very important. Additionally, you should present the results of the chemical analysis of the bars in T6 Temper.
  8. Fig. 7 should be improved, the fonts are too small and should be enlarged.
  9. What form have a primary particle of the Zr–rich phases? How big they are? How Zr is distributed on the cross-section of the grain? What happened with the Zr-rich phase after homogenization?
  10. Page 7, line 191 in Fig. 6 should be Fig. 8
  11. Fig. 11 – Mechanical properties – the results are in the measurement error limit. The differences between DCC and LFEC are very small.
  12. Page 9 line 230: you should not present the results of the UTS and YS increasing with decimal precision.
  13. Page 10, line 270 authors write: “In this study a layer with large size columnar grains in the border of ingot because…..” Fig. 2 didn’t show “large, columnar grains”, there is rather coarse rose-like dendrite grain. Please explain this or show another picture, maybe from the second section.
  14. Conclusion no 4: I do not quite agree with this conclusion – the history of the billets treatment before finally aging probably destroyed the primary formed microstructure – as I mentioned earlier, the mechanical properties are in the measurement error limit.

Author Response

Reviewer: #3

  1. The comments(You didn’t explain, why such conditions for homogenization were selected: 480oC, 36h – the time is so long, the ingots were not big – diameter of 166mm. Probably the shorter time would be also effective and sufficient for ingots homogenization.)

Response: We appreciate for the reviewer’s comment. The 5A90 alloy selected in the experiment has a serious segregation trend. We tried to make the alloy elements fully diffused by increasing the homogenization time. In addition, it has been reported that the homogenization heat treatment of the alloy at 475℃ for 48 hours can make the distribution of Mg uniform, and the alloy can obtain the best comprehensive performance. http://www.cnki.com.cn/Article/CJFDTotal-YSKY404.008.htm?UID=WEEvREcwSlJHSldTTEYzVTFPV2k0dVh0eW1qcXFxVjZUYnpuRlNEdGFlMD0%3d%249A4hF_YAuvQ5obgVAqNKPCYcEjKensW4IQMovwHtwkF4VYPoHbKxJw!!&autoLogin=0

  1. The comments(Fig. 4 – it would be better when the relative segregation rate of Mg and Li will be presented on the graphs with the same scale – then it is easier to compare)

Response: We appreciate again for the reviewer’s suggestion. Since the different equilibrium distribution coefficient k of Mg and Li in aluminum (kMg<kLi), it leads to a difference in the degree of segregation. If the same scales be used, the macro-segregation of Li can not be clearly described as the figure shown.

  1. The comments(Line 145, page 5: ci and co should be a bottom line: ci, co)

Response: Thanks a lot for the well-read reviewer. We have revised the clerical error.

  1. The comments(Page 5 – please explain what is the reason of higher segregation tendency of Mg, than Li)

Response: Thank you for your valuable advice. We have added the reason of higher segregation tendency of Mg, than Li.

  1. The comments(Fig. 5 – the standard deviation should be also presented on the graph of hardness distribution) 

Response: We appreciate for the reviewer’s valuable advice. We have added the error bars (standard error value) on the hardness plot.

  1. The comments(The microstructure performed with higher magnification should be presented in the paper (higher, than presented in Fig. 6 and 9) – then the morphology of the eutectic phase and also others phases will be visible and can be described.)

Response: Thank you for your valuable advice. We have replaced the original pictures with high-magnification pictures, and the morphology of the eutectic phase and also others phases have also been marked.

  1. The comments(Why the authors presented only the microstructures of the ingots? Why you didn’t present the microstructure of the bars in T6 temper?  this is also very important. Additionally, you should present the results of the chemical analysis of the bars in T6 Temper.)

Response: Thank you for your valuable advice, we have added the content of the microstructure of the bars in T6 temper. After the homogenizing treatment the ingot with 166 mm was extruded into bars with a diameter of 16 mm by the four-hole mold,

After this process, the chemical composition of the alloy cast by DCC and LFEC is basically is not much different.

  1. The comments(Fig. 7 should be improved, the fonts are too small and should be enlarged.)

Response: We appreciate for the reviewer’s valuable advice. We have improved the figure.

  1. The comments(What form have a primary particle of the Zr–rich phases? How big they are? How Zr is distributed on the cross-section of the grain? What happened with the Zr-rich phase after homogenization?)

Response: We appreciate again for the reviewer’s suggestion. We have added the content of primary Al3Zr particle in P6 line165~190.

  1. The comments(Page 7, line 191 in Fig. 6 should be Fig. 8)

Response: We are sorry for the poor mistake.The error has be revised.

  1. The comments(Fig. 11 – Mechanical properties – the results are in the measurement error limit. The differences between DCC and LFEC are very small.)

Response: We appreciate again for the reviewer’s comment. After applying the LFEC the average ultimate tensile strength and yield strength of the as-aged alloys both increase with about 30MPa. The tensile test was repeated five times to ensure the reliability of the data, and then the average values of the yield strength (YS), ultimate tensile strength (UTS), and elongation (EL) were calculated.  Both the average value and standard error can indicate the effect of LFEC on the mechanical properties of the alloy.

12 The comments (Page 9 line 230: you should not present the results of the UTS and YS increasing with decimal precision.)

Response: We have revised the presentation about he results of the UTS and YS.

13 The comments (Page 10, line 270 authors write: “In this study a layer with large size columnar grains in the border of ingot because…..” Fig. 2 didn’t show “large, columnar grains”, there is rather coarse rose-like dendrite grain. Please explain this or show another picture, maybe from the second section)

Response: Thanks a lot for the well-read and careful reviewer. We have revised the presentation about dendrite grain. Because of the observation angle chosen by this research is perpendicular to the cross section, it cannot be displayed columnar grains in the border of ingot. We admit that using the expression "columnar grains" may confuse the readers, so we removed this expression.

14 The comments (Conclusion no 4: I do not quite agree with this conclusion – the history of the billets treatment before finally aging probably destroyed the primary formed microstructure – as I mentioned earlier, the mechanical properties are in the measurement error limit)

Response: We appreciate again for the reviewer’s comment. We can see from Figure 11 that the refinement effect of LFEC on the microstructure could be inherited from casting stage to aging stage. The role of grain refinement is not in doubt. Moreover, an increase in the number of Mg and Li atoms within grains enhances supersaturation in the matrix, which provides the necessary conditions for the precipitation of the strengthening phase. The relevant explanations are supplemented at the end of Part 4.

We appreciate for editor Zhang and reviewers’ warm work , and hope that the correction will meet with approval. We will never give up the chance to improve the quality of our manuscript. 

Round 2

Reviewer 3 Report

Thank you for taking my comments into account, I accept the article in the present form.